# Mindfulness and Smartphone Addiction before Going to Sleep among College Students: The Mediating Roles of Self-Control and Rumination

**Shi-Shi Cheng [1], Chun-Qing Zhang [1,2] and Jiang-Qiu Wu [3,*]**

1   Department of Sport, Physical Education and Health, Hong Kong Baptist University, Hong Kong, China; shischeng2@hkbu.edu.hk (S.-S.C.); cqzhang@hkbu.edu.hk (C.-Q.Z.)
2   Department of Psychology, Sun Yat-Sen University, Guangzhou 510275, China
3   Department of Business Administration, Guangdong University of Technology, Guangzhou 510006, China
*   Correspondence: jqiu_woo@gdut.edu.cn

**Abstract:** This study aims to examine the effects among college students of mindfulness on smartphone addiction before going to bed at night. We examined the mediating roles of self-control and rumination on the mindfulness–smartphone addiction path. Participants ($n$ = 270, 59.3% females, 18–24 years old) completed self-reporting questionnaires measuring mindfulness, self-control, smartphone addiction, and rumination. In addition to the correlation analysis, we adopted a stepwise regression analysis with bootstrapping to test the mediating effects. It was found that mindfulness was inversely related to smartphone addiction before going to sleep. Most importantly, self-control and rumination significantly mediated the effects of mindfulness on smartphone addiction among college students. The findings of this study indicated that mindfulness training is beneficial to improve the ability of self-control and reduce rumination levels, thereby inhibiting the negative impact of smartphone addiction on college students before they go to sleep, and further promoting their sleep health and mental health.

**Keywords:** mindfulness; smartphone addiction before going to sleep; self-control; rumination; college students

## 1. Introduction

The internet, with smartphones as its main carrier, has substantially influenced the daily lives of people around the world, with the rapid development of information technology around the world. Consequently, people have become increasingly inseparable from their smartphones and have become "phubbers", unwittingly suffering from "smartphone addiction" [1]. Addiction can be defined as a chronic and elementary disease that is involved in brain motivation and memory in a continuously developing process that could become a fatal disease if not treated promptly [2]. Shin and Dey [3] believe smartphone addicts have to carry their smartphones at all times and check them frequently, and they set several indicators of the addiction assessment. Examples of these indicators include an online time of over five hours every day, addicts decreasing their interest in other activities, and having less social interaction with others [4]. In a recent survey, 66% of individuals who participated in the survey showed signs of nomophobia and touched their phones on average 2617 times a day [5]. In the USA, a recent study indicated that U.S. teens spent on average nine hours on the internet per day, and one out of every two teens felt addicted to his or her device [6]. In China, a survey found that 28% of Chinese netizens were students, which was the largest demographic, and approximately 92% of them had a habit of using their smartphone immediately before going to sleep [7]. Although existing

research indicates that the use of smartphones before going to sleep can greatly reduce sleep quality [8] and may also have a negative impact on the academic performance and interpersonal relationships of undergraduates [9], there are still few intervening measures for smartphone addiction in existing research, and its intrinsic influence mechanisms are even rarer.

Some researchers believe that smartphone addiction is an impulse control disorder [10] and a failure of self-control. Previous research results indicate that self-control is inversely related to smartphone addiction [11,12]. For college students, higher levels of self-control predict lower levels of internet addiction [13]. Rumination—a type of undesirable reaction—is closely associated with mental health problems such as depression, anxiety, and suicide attempts [14]. Additionally, some studies have found that rumination is directly positively associated with internet addiction [15,16]. Thus, we can speculate that self-control and rumination are both proximal factors that contribute to smartphone addiction before going to sleep.

Mindfulness is viewed as a self-regulating method, which emphasizes consciously perceiving and focusing attention on the present. It is associated with enhanced emotional regulation and positive emotional outcomes [17,18], while rumination is related to reduced well-being and also depression, anxiety, and problems of emotional regulation [19–21]. Therefore, mindfulness stimulates the promotion of positive emotions and the maintenance of the best state of emotional and internal calm. Recently, the positive role of mindfulness in providing effective strategies for suppressing addiction problems and psychological problems has attracted the attention of many scholars. Mindfulness therapy has been used to treat a variety of mental disorders, including behavioral addiction [22,23]. One of the most frequently studied areas is pathological gambling therapy based on mindfulness [24]. This method has also been used to treat workaholism [25] and sexual addiction [26]. Additionally, mindfulness is also considered as a method suitable for treating smartphone addiction [27–29]. Recent research has found that the effect of nomophobia on problematic smartphone use weakens as mindfulness increases [30]. Certain mindfulness training methods can reduce withdrawal symptoms and incidences of relapse, regulate the emotional state of anxiety associated with smartphone addiction, and even help addicts identify the intrinsic value of life and control occurrences of rumination. However, there is little empirical research and insufficient theoretical contributions to the intervention effect of mindfulness on smartphone addiction. There are also relatively few studies on the level of mindfulness in China affecting the behavior of certain people through rumination and self-control. Therefore, we concluded that self-control and rumination might mediate the effects of mindfulness on late-night smartphone use, and we analyzed the pathways of self-control and rumination as the after-effects of mindfulness and the antecedents of smartphone addiction in this study. Although Mandal et al. [31] have proposed that emotional regulation and self-control may be mediating variables in mindfulness and smartphone addiction, more empirical research is needed. Testing the mediating modes of mindfulness can improve the intervention methods of preventing and reducing the negative effects of smartphone addiction by increasing college students' mindfulness and self-control levels and reducing their ruminations.

In order to overcome the research gap previously mentioned, this study examined a model proposing that self-control and rumination mediated the effects of college students' mindfulness on smartphone addiction before going to sleep at night (see Figure 1). Specifically, we have three hypotheses:

**Hypothesis 1 (H1).** *Mindfulness results in a significant reduction in smartphone addiction before going to sleep at night.*

**Hypothesis 2 (H2).** *Self-control mediates the relationship between mindfulness and smartphone addiction before going to sleep at night.*

**Hypothesis 3 (H3).** *Rumination mediates the relationship between mindfulness and smartphone addiction before going to sleep at night.*

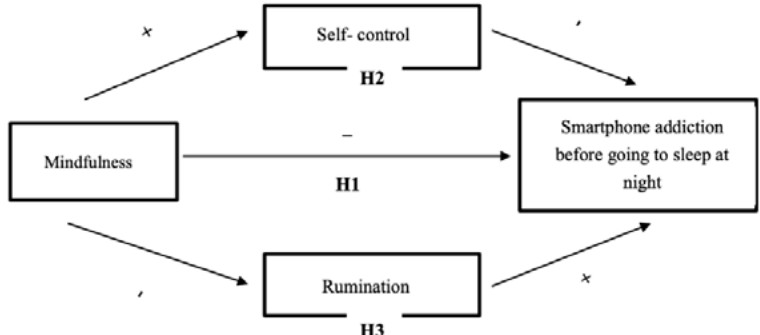

**Figure 1.** The mediation model from mindfulness to smartphone addiction via self-control and rumination.

## 2. Results

### 2.1. Descriptive Statistics

Before performing the hypothesis tests, we conducted Harman's single factor test to investigate issues of common method variance. The first factor accounted for 30.3% of the variance. Since no single factor emerged and the percent variance explained by the method factor was less than the method factor cutoff value of 40% [32], we concluded that the common method variance was not a significant determinant and would not negatively affect the results.

Table 1 shows the means, standard deviations, and intercorrelations of the variables. It demonstrates that mindfulness and smartphone addiction before going to sleep at night was clearly negatively correlated ($r = -0.48$, $p < 0.01$). Moreover, mindfulness was negatively correlated with negative emotions ($r = -0.59$, $p < 0.01$), and positively correlated with self-control ($r = 0.60$, $p < 0.01$). Furthermore, self-control ability was negatively correlated with late-night smartphone use ($r = -0.67$, $p < 0.01$) and negative emotions were positively correlated with smartphone addiction before going to sleep ($r = 0.49$, $p < 0.01$).

**Table 1.** Means, standard deviations and correlations among the key study variables.

| Variable | M | SD | 1 | 2 | 3 | 4 | 5 | 6 |
|---|---|---|---|---|---|---|---|---|
| 1. Gender | 1.57 | 0.50 | - | | | | | |
| 2. Grade | 2.69 | 1.29 | −0.08 | - | | | | |
| 3. Mindfulness | 46.39 | 9.55 | −0.05 | −0.01 | - | | | |
| 4. Smartphone Addiction | 17.61 | 4.89 | −0.01 | 0.11 | −0.48 ** | - | | |
| 5. Self-control | 55.77 | 9.41 | 0.05 | −0.05 | 0.44 ** | −0.67 ** | - | |
| 6. Rumination | 53.92 | 12.60 | −0.01 | 0.05 | −0.60 ** | 0.49 ** | −0.58 ** | - |

Note: ** $p < 0.01$.

According to the correlation analysis, we also found that two variables: smartphone addiction time and time of going to sleep were significantly positively correlated with smartphone use before going to sleep. Furthermore, the statistical data also demonstrated that nearly 75% of college students used smartphones for more than 30 min before going to sleep, and about half of the respondents had gone to sleep after midnight. This illustrated that late-night smartphone use had a great impact on college students' sleep patterns and sleep quality.

### 2.2. Stepwise Regression Analysis

In this research, we regarded self-control and negative emotions as mediating variables between mindfulness and smartphone use before going to sleep at night. The testing of mediating variables by multiple linear regressions, could be divided into three specific steps: (1) test whether the independent variable affects the mediating variable; (2) test whether the independent variable affects the dependent

variable; (3) when the mediating variable is considered, does the effect of the independent variable on the dependent variable disappear or significantly reduce?

　　As shown in Table 2, we firstly tested mindfulness as the independent variable. It had a significantly negative relationship with smartphone addiction before going to sleep as the dependent variable ($\beta = -0.48$, $p < 0.001$) and the $R^2$ was 24%. Next, after entering both mindfulness and self-control as independent variables into the regression equation, the causal relationship between mindfulness and smartphone addiction before going to sleep became insignificant. However, the effect of the mediating variable (self-control) on the dependent variable (smartphone addiction before going to sleep) existed ($\beta = -0.23$, $p < 0.001$), and the $R^2$ increased from 24% to 50%. Therefore, it can be seen that using both mindfulness and self-control simultaneously as the independent variables can better forecast smartphone addiction before going to sleep.

**Table 2.** The mediating effect of self-control on the path from mindfulness to smartphone addiction before going to sleep.

| Variables | Model 1 | Model 2 | Model 3 | |
|---|---|---|---|---|
| | Self-Control | Smartphone Addiction | Step 1 | Step 2 |
| Control variables | | | | |
| Gender | 0.07 | 0.02 | −0.03 | 0.01 |
| Grade | −0.04 | 0.08 | 0.10 | 0.08 |
| Independent variable | | | | |
| Mindfulness | 0.44 *** | - | −0.48 *** | −0.23 * |
| Mediating variables | | | | |
| Self-control | - | −0.67 *** | - | −0.57 *** |
| F | 22.42 | 77.84 | 28.63 | 68.50 |
| R | 0.45 | 0.68 | 0.49 | 0.71 |
| $R^2$ | 0.20 | 0.46 | 0.24 | 0.50 |
| Adj. $R^2$ | 0.19 | 0.46 | 0.23 | 0.50 |

Note: * $p < 0.05$; *** $p < 0.001$.

　　Additionally, consistent with the mediating effect of self-control, both mindfulness and rumination as independent variables can better predict smartphone addiction before going to sleep than mindfulness alone ($R^2$ increased from 24% to 30%) (see Table 3). Thus, we conclude that negative emotions mediate the influence of mindfulness on late-night smartphone addiction.

**Table 3.** The mediating effect of rumination on the path from mindfulness to smartphone addiction before going to sleep.

| Variables | Model 1 | Model 2 | Model 3 | |
|---|---|---|---|---|
| | Negative Emotions | Smartphone Addiction | Step 1 | Step 2 |
| Control variables | | | | |
| Gender | −0.05 | −0.01 | −0.03 | −0.02 |
| Grade | 0.01 | 0.09 | 0.10 | 0.09 |
| Independent variable | | | | |
| Mindfulness | −0.60 *** | - | −0.48 *** | −0.29 ** |
| Mediating variable | | | | |
| Ruminations | - | 0.49 *** | - | 0.31 ** |
| F | 52.59 | 29.73 | 28.63 | 29.12 |
| R | 0.61 | 0.50 | 0.49 | 0.55 |
| $R^2$ | 0.37 | 0.25 | 0.24 | 0.30 |
| Adj. $R^2$ | 0.36 | 0.24 | 0.23 | 0.29 |

Note: ** $p < 0.01$, *** $p < 0.001$.

　　Based on the bootstrap test, the results showed that self-control and rumination played significant mediating roles in the influence of college students' mindfulness on smartphone addiction prior to going to sleep at night. The indirect effects did not include zero in the 95% bootstrap confidence interval, which proved that the indirect effects were significant and the mediating effects existed (see Table 4).

**Table 4.** Bootstrap tests of the mediating effects of rumination and self-control on the path from mindfulness to smartphone addiction before going to sleep.

| Paths | 95% Bootstrap CI | | | |
|---|---|---|---|---|
| | **Effect** | **SE** | **LL** | **UL** |
| Mindfulness → Self-control → Smartphone Addiction before Going to sleep | | | | |
| Indirect effect | −0.13 | 0.02 | −0.17 | −0.08 |
| Total effect | −0.12 | 0.03 | −0.16 | −0.06 |
| Mindfulness → Rumination → Smartphone Addiction before Going to sleep | | | | |
| Indirect effect | −0.10 | 0.03 | −0.15 | −0.05 |
| Total effect | −0.15 | 0.03 | −0.21 | −0.08 |

Note: CI = confidence interval; SE = standard error; LL = lower limit; UL = upper limit.

## 3. Discussion

This study aimed to investigate the effect of Chinese college students' mindfulness levels on smartphone addiction before going to sleep at night. Based on the regression analysis, we confirmed the hypothesized mediating roles of self-control and rumination. Findings on the mediating roles of self-control and rumination informed the changing mechanism of a mindfulness intervention on preventing late-night smartphone addiction.

### 3.1. Theoretical Implications

From the correlation analysis and regression analysis of mindfulness and smartphone addiction before going to sleep, we believe that mindfulness shows a significant negative correlation with smartphone addiction late at night. It is consistent with previous research results, suggesting that undergraduates with higher levels of mindfulness have stronger self-control capabilities and lower levels of rumination, as well as lower levels of smartphone addiction before going to sleep. This may further demonstrate that mindfulness traits have positive effects on the self-development and psychosocial adaptation of individuals, and even play protective roles in the rumination affecting an individual's physical and mental health [33]. Indeed, from the perspective of the nature of mindfulness, undergraduates who possess higher mindfulness levels can easily focus on their studies, instead of dwelling on the past and worrying about the future. This thereby reduces ruminations associated with smartphone addiction and improves the undergraduates' sleep quality. Overall, this research further clarifies the positive influence of mindfulness mechanisms, supplements the research on the inhibiting effect of mindfulness on smartphone addiction, and expands the study on the mediating effects of mindfulness on addictive behavior.

Furthermore, this research is also consistent with the existing research on both "The Positive Emotion-Building Theory" [34] and "The Relationship between Emotions and Sleep Quality" [35]. The findings of our study indicate that self-control and rumination mediate the effects of mindfulness on smartphone addiction before going to sleep. It demonstrates that self-control factors and emotional factors are proximal factors that affect sleep quality. It also paves the way for future studies to explore more mediating factors that may play a bridging role between mindfulness and smartphone addiction, to fully reveal how mindfulness specifically affects smartphone addiction. Additionally, the research method of this study for testing mediating roles will also help to form a stable experimental paradigm of the mindfulness mechanism.

### 3.2. Practical Implications

According to our findings, there are several important practical implications. Firstly, we believe that it is essential for undergraduates to improve their emotional control abilities via mindfulness training. Fredrickson [36] combined the expansion and construction theory of positive emotions with mindfulness training and believed that mindfulness training could increase the positive emotions of trainees. Moreover, Arch and Craske [37] found that members of a mindfulness intervention

breathing group did not feel very distressed when watching slides that triggered negative emotional experiences, and that mindfulness training played an important role in adapting to negative emotional stimuli. Additionally, some scholars argue that effectively implemented mindfulness therapy may, over time, lead to the dampening and eventual elimination of addiction. Thus, colleges and parents should encourage and guide undergraduates via mindfulness training to actively and optimistically face the present and repel rumination. By eliminating negative emotions and developing a healthy mentality, students can better reduce the degree of late-night smartphone use, thereby improving their sleep quality.

Secondly, through exploring the influence of smartphone addiction on college students' behavior and sleep quality, this study also has some practical implications on promoting the physical and mental health development of college students. Previous studies and clinical trials [38] have shown that sleep deprivation can lead to endocrine and emotional disorders, and even an increased risk of cardiovascular disease and cancer. Therefore, it is essential to improve undergraduates' self-control through mindfulness training to control the occurrence of smartphone addiction late at night.

Several approaches can help to control addictive behavior associated with smartphones. Firstly, colleges should cultivate the healthy personal development of undergraduates and build a harmonious group atmosphere through positive propaganda and education to advocate rational smartphone use. Secondly, colleges should organize a variety of activities to reduce loneliness and rumination of college students, reducing their interpersonal anxiety and enhancing their sense of belonging. Moreover, colleges should also run courses for undergraduates on mindfulness training to promote emotional regulation and alleviate smartphone addiction in undergraduates. More importantly, for undergraduates themselves, they should actively participate in campus activities and face-to-face conversations with friends. Perhaps other activities such as going traveling from time to time are good ways to regulate rumination.

### 3.3. Limitations and Future Research

Several limitations of this research should be emphasized, providing new directions for future exploration. Firstly, it should be noted that the college students who participated in the questionnaire mostly came from Mainland China. Due to the limited sample size, they are not representative of all young people and may have been influenced by unique cultural and social contextual factors. Therefore, our future research may select sample surveys from a wider population from different cultures, such as a transnational survey. Moreover, other factors such as age, gender, personality, family impact on income, and the environment, may also have had an impact on the results of the study. Therefore, we suggest that further investigation and construction of more complex theoretical models and a more comprehensive discussion of the relationships among these issues are needed in future research. Additionally, as for research methods, we suggest that experimental research methods, psychological intervention research methods, and interview research methods can be adopted to further verify the data.

## 4. Methods

### 4.1. Procedure

In this research, we adopted a two-wave survey study with a time lag of six weeks. We used an online survey of college students from eight universities in the province of Guangdong, China. Inclusion criteria were that participants (a) should have a smartphone and use it regularly and (b) were willing to respond to a second follow-up survey. We contacted teachers in the student affairs offices of the eight universities to recruit participants and collected contact information for the students who had volunteered to participate. In the first wave, we distributed questionnaires on the internet, mainly measuring independent variables (dispositional mindfulness) and mediating variables (self-control and rumination levels). We also asked participants to leave contact information such as mobile phone

numbers and email addresses. We approached 307 college students, of whom 294 submitted responses with valid data at this stage. About one month later, the second-wave online questionnaire was distributed from the collected samples according to the contact information left by the participants. We measured result variables (smartphone addiction before going to sleep), and control variables (gender, grade) as well as two related variables (time of going to sleep and time spent using smartphones before going to sleep). Ultimately, 270 participants provided responses at this stage, with a sample retention ratio of approximately 87.9% (see Table 5 for the demographic information).

**Table 5.** Demographic details for the final sample ($N = 270$).

|  | *N* | **Frequency** |
|---|---|---|
| Gender | | |
| Male | 110 | 40.7% |
| Female | 160 | 59.3% |
| Grade | | |
| Freshman | 84 | 31.3% |
| Sophomore | 34 | 12.6% |
| Junior | 59 | 21.9% |
| Senior | 93 | 34.4% |
| Universities | | |
| Guangdong University of Technology | 140 | 51.9% |
| Sun Yat-sen University | 16 | 5.9% |
| Jinan University | 26 | 9.6% |
| South China Normal University | 24 | 8.9% |
| Guangzhou University | 16 | 5.9% |
| Guangdong University of Finance and Economics | 15 | 5.6% |
| South China Agricultural University | 16 | 5.9% |
| Guangdong University of Finance | 17 | 6.3% |

*4.2. Measures*

All English-based materials were translated into Chinese according to the "translation/ back-translation" procedure, and a Likert-type scale ranging from 1 (strongly disagree) to 5 (strongly agree) was used for measuring mindfulness, smartphone addiction before going to sleep, self-control, and rumination.

Mindfulness. Mindfulness was measured with the five facet mindfulness questionnaire (FFMQ) (Baer [39]). The entire mindfulness scale consists of 39 items with five dimensions (observation, description, action with awareness, non-judgmental intrinsic experience, non-reactivity to intrinsic experience). Of these five dimensions, we selected the two most relevant to smartphone addictive behavior (action with awareness and non-judgmental intrinsic experience) with 16 items to measure mindfulness. In this research, action with awareness was defined as the inability to focus on the current situation and thus use the smartphones constantly. The non-judgmental intrinsic experience was defined as acts that are excessively concerned with internal experience and cannot be dissociated. Sample items include "I am easily distracted when working or studying". We adopted a Chinese version of the FFMQ translated by Deng et al. [40]. Items were rated on a five-point Likert scale from 1 (strongly disagree) to 5 (strongly agree). The Chinese version of the FFMQ in this study was consistent with the international version and was shown to be reliable ($\alpha = 0.88$).

Smartphone addiction before going to sleep at night. This was measured by the mobile phone addiction tendency scale (Xiong [41]). This scale has six items measuring three dimensions: withdrawal symptoms, prominent behaviors, and mood changes. Sample items included "I concentrate on smartphone content which delays my going to sleep". Items were rated on a five-point Likert scale from 1 (strongly disagree) to 5 (strongly agree). The scale in this study satisfied international standards and was shown to be reliable ($\alpha = 0.84$).

Self-control. The self-control scale (SCS) (Tangney [42]) involved more comprehensive behavioral measures such as the regulation of consciousness, inhibition of impulse, change of emotion, and change of habit. The Chinese version of the SCS was translated and revised by Tan [43] with 19 items for Chinese college students. After reliability testing, we retained 18 items related to the subject of this research, and excluded the item "I spend too much money". Sample items included "I can control the amount of time I spend on my smartphone before going to sleep". Items were rated on a five-point Likert scale from 1 (strongly disagree) to 5 (strongly agree). The Chinese version of the SCS scale in this study was consistent with the international version and was demonstrated to be reliable ($\alpha = 0.85$).

Rumination. We used the 22-item ruminative responses scale (RRS) (Nolen-Hoesksema [44]) to measure rumination. Based on the theory of response mode of depression, rumination may be a predisposing factor for negative emotional attacks (e.g., depression) [45]. The Chinese version of the RRS was first translated into Chinese by two clinical psychologists, back into English, and then into Chinese again by two linguists at McGill University in Canada. The scale consists of three factors: depression, brooding, and reflection. Based on research needs, we selected 18 items related to late-night smartphone use for college students. Sample items included "I often think about how lonely I am". Items were rated on a five-point Likert scale from 1 (strongly disagree) to 5 (strongly agree). The Chinese version of the RRS scale in this study met international standards and passed a reliability test ($\alpha = 0.94$).

To better test the hypothesis, we also set a control variable (age and grade). Moreover, this study also added two covariates in the correlation analysis: time spent using smartphones and the time of going to sleep. This deeply explored the substantial impact of smartphone addiction on undergraduates' behavior before they went to sleep, and provided some practical implications for their physical and mental health.

### 4.3. Data Analysis

IBM SPSS Statistics 23.0 was used for the descriptive analysis, correlation analysis, and multiple linear regression analysis to explore the degree of correlation between mindfulness, self-control, rumination, and smartphone addiction before going to sleep at night. It also tested whether self-control and rumination mediated the effect of mindfulness on late-night smartphone addiction. Additionally, we also used Model 4 of the bootstrapping approach suggested by Preacher and Hayes [46] to further verify the significance of the mediating effects. A method of repeated random sampling was adopted in this part to determine whether the mediating effect was significant or not according to whether the 95% bootstrap confidence interval contained zero.

## 5. Conclusions

The current study aimed to investigate the effect of undergraduates' mindfulness levels on smartphone addiction before they went to sleep at night under the two mediating effects of self-control and rumination. Findings showed that the higher the mindfulness level of undergraduates, the stronger their self-control ability and the lower the level of rumination, as well as the lower the degree of smartphone addiction before going to bed. In addition, both self-control and rumination played mediating roles in the influence of mindfulness on smartphone addiction late at night. Finally, the findings of the current study demonstrated that mindfulness is crucial for undergraduates to improve their emotional regulation and promote their physical and mental health. Future studies should further confirm the changing mechanisms of mindfulness training using randomized controlled trials.

**Author Contributions:** Conceptualization, J.-Q.W.; data curation, S.-S.C.; formal analysis, S.-S.C.; investigation, S.-S.C.; methodology, S.-S.C.; supervision, J.-Q.W.; writing—original draft, S.-S.C.; writing—review and editing, C.-Q.Z. All authors have read and agreed to the published version of the manuscript.

**Funding:** This research received no external funding.

**Conflicts of Interest:** The authors declare no conflict of interest.

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
