# Peer review of "Mindfulness and Smartphone Addiction before Going to Sleep among College Students: The Mediating Roles of Self-Control and Rumination"

_2624-5175, doi:10.3390/clockssleep2030026_

Round 1
Reviewer 1 Report
- Lines 6,7, 8 - the authors must complete with necessary data
- First question is why the authors choose only girls in this study, what is the explanation? The authors must detailed this
- Introduction - the number of references is not correct, please see the template instructions of journal. Also, why the authors not presented others studies from other countries, just from China is enough?
- I think a table with the descriptive data of respondents is necessary, please include in subsection Methods these variables: gender, years, age, universities (eight) etc.
- The evaluation tools are presented correctly and completely
- From Figure 1 the authors must put point after title of these.
- Lines 191-200 - please rearrange the paragraphs in text, there are to much space between sentences
- Please insert more current studies related to the topic,
- Once again the authors must change the citations in all paper !!!!!!!
- References is not in accordance with the policy of journal.

Author Response
Dear Reviewer,
We appreciate the time and effort you have spent in reviewing our manuscript (ID: clockssleep-902412). Your comments are really thoughtful and helpful. Thus, we have revised the manuscript according to your kind advices and detailed suggestions. We highlighted the changes in red color.
Please find our responses to reviewers’ comments and suggestions:
Comment 1: First question is why the authors choose only girls in this study, what is the explanation? The authors must detail this.
Response 1: Thank you for this comment. Actually, we recruited both male and female as our study samples (n = 277, 59.2% females, 40.8% male). Please refer to Line 13 and Line 174.
Comment 2: The number of references is not correct, please see the template instructions of journal. Also, why the authors not presented others studies from other countries, just from China is enough?
Response 2: Thank you for this comment. We have reformatted the references and the number of references via Endnote according to your comments. Given that our research focused on Chinese college students, we introduced the current situation of Chinese college students’ smartphone addiction at the beginning of the introduction. We acknowledge that smartphone addiction is a worldwide phenomenon. Therefore, we have supplemented five studies about the status quo of other countries that are related to the topic of our study (see Line 37-50, Line 55 and Line 57 or see Reference 5, 6, 12, 13, 18).
Comment 3: I think a table with the descriptive data of respondents is necessary, please include in subsection.
Response 3: Thank you for your suggestions. We have supplemented a table with the descriptive data of respondents (Table 1. Demographic Details for the whole sample) in line 174-175, including gender, grade and universities.
Comment 4: From Figure 1 the authors must put point after title of these.
Response 4: Thank you for your advice. We have included point after title of Figure 1 in line 165.
Comment 5: Lines 191-200 - please rearrange the paragraphs in text, there are too much space between sentences.
Response 5: Thank you for your suggestion. We have adjusted the texts of Lines 191-200. Please refer to Lines 203-212 of the revised manuscript.
Comment 6: Please insert more current studies related to the topic.
Response 6: Thank you for this suggestion. We have supplemented five recent studies related to our research topics based on your comments (see Line 37-50, Line 55 and Line 57 or see Reference 5, 6, 12, 13, 18).
Comment 7: Once again the authors must change the citations in all paper!!!!!!! References is not in accordance with the policy of journal.
Response 7: Thank you for this comment. We have checked and fixed the citation issue via Endnote. We hope this is now in line with the policy of the journal.
Reviewer 2 Report
Well written paper. I have no specific suggestions regarding the paper.
The method is appropriate (but now after another reading I am thinking that maybe there is a hindrance of a small N of participants and it could be further detailed how they were recruited); it is true though that studied concepts are very much related.
the paper has a practical side to it, as it highlights not only the importance of mindfulness, but mechanisms behind it, which many other studies lack. So this gives it a better chance for the wider audience.
Just a minor comment:
Mindfulness has become a self-regulating method --> probably is a self-regulating method?
Author Response
Dear Reviewer,
We appreciate the time and effort you have spent in reviewing our manuscript (ID: clockssleep-902412). Your comments are really thoughtful and helpful. Thus, we have revised the manuscript according to your kind advices and detailed suggestions. We highlighted the changes in red color.
Please find our responses to reviewer’ comments and suggestions:
Comment 1: I am thinking that maybe there is a hindrance of a small N of participants and it could be further detailed how they were recruited.
Response 1: Thank you for your suggestions. Indeed, it is one of the limitations of this study that the sample size is not large enough. Actually, we have mentioned it at section 4.3 (see Line 293-304) "due to the limited sample size limitations and further research, they are not representative of all young people and may have been influenced by unique cultural and social contextual factors. Therefore, our future research may select sample surveys from a wider population from different cultures, such as a transnational survey".
In addition, we elaborated the way we recruited participants. We contacted teachers in the student affairs office of the eight universities in Guangdong Province of China to recruit participants and collected contact information for the students who volunteered to participate (see Line 96-98).

Round 2
Reviewer 1 Report
See attach

Author Response
Dear Reviewers,
We appreciate the time and effort you have spent in reviewing our manuscript (ID: clockssleep-902412). Your comments are really thoughtful and helpful. Thus, we have revised the manuscript according to your kind advices and detailed suggestions. We highlighted the changes in red color.
Please find our responses to reviewer’ comments and suggestions:
Comment 1: For table 1 I have a question: What represent “1, 2, 3, 4” in this table?
Response 1: Thank you for this comment. “1, 2, 3, 4” in Table 1 represent Freshman, Sophomore, Junior and Senior respectively. Please refer to Note. in Line 92.
Comment 2: Figure 1 - please put point after title, not bold the title?
Response 2: Thank you for your advice. We have included point after title of Figure 1 in line 80.
Comment 3: The citations in the text is not correct, see template of journal, must be in square brackets, e.g. [1], or [2,3], or [4-6] etc, see Instructions for Authors, also the References from end of article, please rearrange correct.
Response 3: Thank you for your comments. We have referred to the Instructions for Authors to put the references of text and references from end of article in square brackets. We hope this is now in line with the policy of the journal.
Comment 4: Again, you have space big space between words: see lines 119, 127,129, 180 etc.
Response 4: Thank you. Changes have been made.
Comment 5: Point after title of table 1.
Response 5: Thank you for your advice. We have included point after title of Table 1 in line 92.
Comment 6: At final before references put the ''Author Contributions''.
Response 6: Thank you for this suggestion. We have supplemented ''Author Contributions'' at final before references (see Lines 308-310).